# Regulation of the Concentration Heterogeneity and Thermal Expansion Coefficient in the Metastable Invar FeNi_31.1_ Alloy

**DOI:** 10.3390/ma15238627

**Published:** 2022-12-02

**Authors:** Valery Shabashov, Victor Sagaradze, Andrey Zamatovskii, Kirill Kozlov, Natalya Kataeva, Sergey Danilov

**Affiliations:** M.N. Mikheev Institute of Metal Physics, Ural Branch, Russian Academy of Sciences, 620108 Ekaterinburg, Russia

**Keywords:** Fe–Ni, Invar, thermal expansion coefficient, phase transformations nanostructure, Mössbauer spectroscopy

## Abstract

Mössbauer spectroscopy and electron microscopy study of the active redistribution of Ni atoms during the process of polymorphous transformation α→γ in the metastable FeNi_31.1_ alloy revealed that slow heating (at the rate of 0.2 K/min) results in the depletion of the initial α-phase with a beneficiation of developing disperse γ-phase plates according to the equilibrium diagram. A regulation possibility of the concentration heterogeneity and austenite thermal expansion coefficient resulted from the polymorphous transformation α→γ was shown. Comparison with data of FeNi_35_ alloy irradiation by high-energy electrons responsible for the variation of atomic distribution and thermal expansion coefficient (owing to the spinodal decomposition) was performed.

## 1. Introduction

For the time being, the common theory about the invar effect in alloys is missing. However, it is accepted that the invar properties of alloys are of a magnetic nature [1,2,3,4]. One of the models accounting for the invar anomalies is based on the contribution of structural heterogeneities in these alloys. This fact is proved by a strong dependence of the invar alloy peculiarities on the artificially created heterogeneities [5,6] or plastic deformation [7,8,9]. Models associating the invar properties with concentration heterogeneities are built either on statistical fluctuations [10,11] or on local environment effects [6,12]. Nevertheless, both cases deal with magnetic inhomogeneities related to the composition fluctuations. An example of atomic redistribution variation which result in the invar properties’ disturbance is development of a short- and long-range ordering in the Fe–Ni alloy structure [6,13]. In particular, the thermal expansion coefficient (TEC) growth in the Invar FeNi_35_ alloy is observed after the high-energy electron and ion irradiation due to the short-range ordering development [6,14,15]. The concentration heterogeneity in alloys can be controlled in terms of deformation and thermal-induced structure-phase transformations [16,17,18,19,20]. For instance, metastable Fe–Ni alloys from the Invar range of 30–34 at.% of Ni appeared to have an active Ni redistribution under the heating rate variation in terms of polymorphic α→γ transformation with a further development of austenite with a concentration heterogeneity [19]. The fast heating (200 K/min) of α-martensite in the FeNi_32_ alloy results in transformation with coarse plates’ development in γ-phase via reconstitution of the initial austenite orientation without composition variation. The slow heating (0.2–0.4 K/min) of α-martensite in the FeNi_32_ alloy leads to the transformation in terms of disperse (nanosized) austenite formation with multiple orientations of γ-crystals. The disperse austenite formation results in the development of extended surface of interface α/γ borders where in the range of α→γ transformation active Ni ions redistribution occurs between α- and γ-phases [19]. The current study is aimed to investigate the possibilities of concentration heterogeneity regulation in the Fe–Ni austenite in terms of the polymorphic α→γ transformation that allows us to vary the TEC value of alloy in the wide range. In addition, it was interesting to match the obtained results with ones including the influence of stratification caused by the electron irradiation on TEC.

## 2. Materials and Methods

Binary Fe–Ni alloys with Ni content of 31.1 and 35 at.% were chosen for the study (C ~ 0.04 at.%). The FeNi_31.1_ alloy is a metastable one and tends to form ~70% of martensite in terms of quenching in the liquid nitrogen.

Studied alloys were melted out in the induction furnace using pure elements then homogenized and hardened in water at 1050 °C with further quenching in the liquid nitrogen to obtain α-martensite structure. The reverse transformation α→γ was performed using plates of 20 × 20 × 1 mm size under the heating up to different temperatures with a rate of 0.2 K/min with further cooling by ambient air to the room temperature. Some samples slowly heated during α→γ transformation were exposed to austenisation by the fast heating up to 600 °C in the salt bath (exposition during 60 s) with further water cooling. Indicated procedure permitted us to keep Ni concentration heterogeneity in the single-phase austenitic condition.

The TEC values of the FeNi_31.1_ alloy after α→γ transformation with a slow heating were measured using the Chevenard dilatometer Typ.DP (Switzerland) in the temperature range from minus 50 to 50 °C. Samples were prepared as rods of 50 mm length and 3 mm diameter. TEC values of initial FeNi_35_ alloy and after electron irradiation were measured using the dilatometer DL-1500RHP by ULVAC-SINKU RIKO (Saito City, Japan) at the heating rate of 2 K/min using foils of 8 × 4.5 mm size and 100 micron thickness.

Electron irradiation with energy of 5 MeV, at 150 °C, was performed using a linear accelerator.

Plate samples for Mössbauer study were mechanically ironed with further electrolytic polishing up to 20 micron thickness. Mössbauer measurements were performed, at the room temperature, in the transmission geometry using the spectrometer MS-1101 and ^57^Co(Cr) source with resonance γ-quanta energy of 14.4 kEv. Calibration procedure was performed at room temperature using α-Fe. Due to the local heterogeneity and complex structure of studied alloys, the distribution *p*(*H*) and spectra of phases were reconstituted using the Distri program within the fitting environment of MS-Tools [21].

Electron microscopy was performed using JEM-200CX microscope (JEOL, Tokyo, Japan).

## 3. The Results of the Experiment

### 3.1. Mössbauer Spectroscopy of the FeNi_31.1_ Alloy

Alloy quenching in the liquid nitrogen leads to the martensitic γ→α transformation resulted in the development of ~70% martensite. Spectra of α and γ phases have significant differences and are not overlapped within the internal effective field scale (Figure 1) [22]. Sextet lines of α-phase in BCC appeared to have some small line broadening due to nickel influence on the effective magnetic field of the resonance iron [23]. The mean value of effective magnetic field of α-phase ${\langle H\rangle }_{\alpha }=\sum H_{i}\cdot p(H_{i})$ has no strong dependence from the Ni content [23,24] and shows more quantitative features. The room-temperature spectra (lower than the Curie point) of γ-phases (FCC) in alloys of the Invar range (30–36 at.% Ni) revealed a six-line shape with a wide distribution of the effective magnetic field values in the range from 0 to 305 kOe [6,25]. Spectra analysis of γ-phase was performed using a physical model of the Fe–Ni Invar with noncollinear magnetic structure [26,27]. The Mössbauer spectra appearance of the Fe–Ni Invar as well as *p(H)* and <*H*>_γ_ distributions strongly depend on the Ni content. An atomic distribution analysis in γ-phase matrix is based on *p(H)* distribution and dependence between heterogeneous hyperfine magnetic field and composition fluctuations [18,25,26,27,28]. Density peaks of *p(H)* with larger field values are associated with austenite regions with a higher Ni content. The central part of Mössbauer spectra of γ-phase in studied FeNi_31.1_ alloy revealed the component with almost zero hyperfine magnetic field value which can be related to paramagnetic and antiferromagnetic regions of a metal matrix with low Ni content (less than 29 at.%) around Fe atoms. For quantitative assessment of Ni distribution in the FCC matrix, both *p(H*) distribution and dependence between <*H*>_γ_ and Ni content are used (see Figure 2) [25,29].

### 3.2. Development of the Austenite with the Concentration Heterogeneity during α→γ Transformation

Mössbauer spectra of initial alloy and one after α→γ transformation were fitted using two-phase model for non-overlapped field regions: (i) γ-phase in the region of 0–305 kOe and (ii) α-phase in the region of 305–350 kOe. The Mössbauer spectrum of γ-phase is represented as a superposition of γ_1_ and γ_2_ phases (similarly to [26,27]) related to the austenite in different magnetic states: (i) γ_1_-phase is represented by the singlet and associated with austenite structural regions in paramagnetic and antiferromagnetic states with the Ni content ≤ 29 at.%; (ii) γ_2_-phase is related to the ferromagnetic component of austenite with acute dependence of the mean effective field value <*H*> from the Ni content [18,25,28,29]. Curve 1, shown in Figure 2, is calculated for binary FCC Fe–Ni alloys with 29–45 at.% of Ni within the whole field range (from 0 to 308 kOe), and curve 2 is calculated after the central component γ_1_ deduction. Extraction of both γ_1_ and γ_2_ phases and *p*(*H*) distribution from the spectrum permits us to perform quantitative evaluation of austenite redistribution with simultaneous development of regions depleted and enriched with Ni in the austenite structure.

Using the *p*(*H*) calculation for the spectrum of martensite in the quenched FeNi_31.1_ alloy, a sextet subspectra of α-phase with <*H*>_α_ = 331 kOe as well as the remaining γ-phase represented by the superposition of the central singlet γ_1_ and sextet γ_2_ with <*H*>_γ_ ~ 144 kOe were reconstituted (Figure 1a). It is clearly seen in Figure 1a that the appearance of the spectrum and *p*(*H*) distribution of remained austenite is similar to that of austenite with the concentration homogeneity. The heating of quenched martensite at the rate of 0.2 K/min up to 400–490 °C results in a ratio variation of Mössbauer phases and related hyperfine parameters (see Table 1 and Figure 1a–d).

During the heating in the primary temperature region from 400 to 470 °C, the integral intensity and mean field <*H*>_γ_ values of the γ_2_ component grow, which indicates the presence of α→γ transformation and Ni content growth in the austenite, according to the <*H*>_γ_ dependence shown in Figure 2. The raised Ni volume and concentration in γ_2_ resulted from decreased integral intensity (volume) of the ferromagnetic sextet related to α-phase. At the same time, <*H*>_α_ of α-phase increases from 331 to 337 kOe, which indicates a Ni content decrease [24,25]. The observed variations in spectra demonstrate the polymorphic transformation accompanied by Ni redistribution between phases with decreased Ni content in α-phase and simultaneously increased Ni content in developed austenite. Observed variations are given in Table 1 and shown in Figure 1a–c. Active Ni redistribution during the stage of α→γ transformation occurs due to disperse γ-crystals formation [19]. According to TEM, data disperse plates (of 10–50 nm thickness) of γ-phase are formed in α-phase in the temperature range of 400–470 °C through bainitic transformation, with all 24 orientations permitted by the martensite-like phase transformation [19] (see Figure 3a). It is disperse γ-crystals and the large surface of α/γ borders that appear to be a condition of the observed significant Ni redistribution. It should be mentioned that the mean value of the effective field of the γ_2_ phase, in the temperature range of 400–470 °C, took place due to the large field values of ~270–310 kOe. In this case, along with the appearance of the intensity peak in *p*(*H*), in the range of high-magnitude fields, one observes the growth in intensity of the peaks corresponding to austenite of the initial composition, with preservation of the ratio of their intensities together with their position on the axis *H*. This means that along with the appearance of the γ phase enriched in nickel, the growth of the amount of austenite takes place without any sufficient enrichment of austenite in nickel. The latter being a result of the α→γ transformation can be a consequence of the growth of coarse grains on the «substrate» of retained austenite. At the end of the region of α→γ transformation (470–490 °C), the globular austenite forms. Globular austenite absorbs the concentrationally heterogeneous mixture of α- and γ-phases, retaining the nickel segregation within the globule. Figure 3b shows the structure of a globular grain of a concentration heterogeneous austenite. The dark banded diffraction contrast within the globule corresponds to the thin-plate areas of the nickel-enriched phase, which was absorbed by the growing globular austenite. The stage of α→γ transformation is distinguished by the intensity growth of the paramagnetic component γ_1_ in the austenite up to 12 and 21% and by the remarkable intensity growth of γ_2_ component (Figure 1d and Table 1). This fact indicates the volume growth of the austenite with Ni content ≤ 29 at.% and with Ni content higher than initial one. Thus, the final stage of α→γ transformation simultaneously demonstrates the volume growth of austenite regions depleted with Ni (γ_1_) and enriched with Ni (γ_2_). According to obtained data related to α, γ_1_, and γ_2_ variation, it is possible to conclude that the paramagnetic component γ_1_ was raised from Ni-depleted α-phase after α→γ transformation (see Figure 1c,d and Table 1).

During the final stage in the temperature range, from 490 to 520 °C, a partial leveling of alloy concentration heterogeneity occurs, which can be recognized by the <*H*>_γ_ decrease in γ_2_-phase (see Figure 1d–e and Table 1). This thermal heating stage is characterized by the transformation way switching towards the massive one with γ-globules development. The latter absorbs heterogeneous regarding the Ni content α + γ structure (see Figure 3c).

To reduce the time needed to level the concentration heterogeneity for the mechanism of globular borders moving, the fast heating up to 600 °C (with 60 s exposure) in the salt bath was performed for samples with the concentration heterogeneity. Mössbauer spectra of samples after described thermal treatment are shown in Figure 4.

The appearance of measured spectra and *p*(*H*) distribution indicates complete α→γ transformation. However, the spectrum and *p*(*H*) distribution of formed austenite revealed qualitative variation from the initial tempered austenite and one after the fast heating of quenched martensite (see Figure 4a). This fact indicates a significant increase in magnetic and concentration heterogeneity of the structure. To evaluate the rate of magnetic and concentration heterogeneity, the correspondent spectra were fitted in terms of the austenite model with the concentration heterogeneity using three components: (i) γ_1_, associated with the austenite in the paramagnetic state with Ni concentration ≤ 29%; (ii) γ_2_, related to the fields’ region of the ferromagnetic austenite with composition close to the initial one; (iii) γ_3,_ with field values related to the austenite with Ni content higher than that in the initial one up to field values correspondent to the FeNi structure ordered in terms of equiatomic composition [6,18]. Different magnetic phases, γ_1_, γ_2_ and γ_3_, were revealed in the austenite spectrum and represented as austenite structural regions with various composition in accordance with obtained Mössbauer data and electron microscopy results. The phases are as follows: (i) γ_1_ is formed instead of Ni-depleted α-phase; (ii) γ_2_ is related to the structure of retained austenite and hardly participate in Ni redistribution processes; (iii) γ_3_ is associated with newly formed austenite developed on the base of Ni-enriched disperse γ-plates. Obtained results shown in Figure 4 and given in Table 2 indicate the possibility to increase the degree of austenite concentration heterogeneity using the fast heating up to 600 °C in the salt bath. In particular, the aforementioned heating, starting from 450 and 470 °C, allows us to obtain the highest degree of concentration heterogeneity.

### 3.3. The Relation between the TEC and Degree of Concentration Heterogeneity in the Austenite

Mössbauer spectroscopy demonstrated that the highest concentration heterogeneity in austenite occurred during the slow heating up to 450–470 °C, in terms of the highest Ni-depletion of α-phase and enrichment of γ-phase in accordance with the equilibrium diagram for Fe–Ni alloys [30]. The fast heating up to 600 °C provide the completion of α→γ transformation with fixed concentrated heterogeneous structure in the form of austenite. TEM revealed the structure with nanosized concentration inhomogeneities (as contrasts) remained instead of the structure obtained by the heating during α→γ transformation (see Figure 3b,c). Observed contrasts can result from different degrees of etch ability for austenite regions with various composition. Mössbauer spectra and *p*(*H*) distributions of the austenite with the concentration heterogeneity obtained during α→γ transformation proved to be similar to that of the austenite obtained using electron irradiation (Figure 5). Atomic distribution histograms for the FeNi_35_ alloy were calculated using the superposition model of paramagnetic γ_1_ and ferromagnetic γ_2_ spectral components associated with γ-phase (Figure 5). Similar spectra reflect the development of bulk structures with Ni distribution significantly different in comparison with that in homogenized Fe–Ni Invars [6,18,29].

It can be expected that the variation of concentration heterogeneity degree in the austenite from studied FeNi_31.1_ alloy (namely, the size of central paramagnetic peak related to γ_1_-phase with ≤29 at.% Ni and component with magnetic field value of 290–305 kOe and 40–50 wt.% Ni) will result in increased TEC value for this alloy. The highest concentration heterogeneity in the austenite is developed during the slow heating up to 450–470 °C, accompanied by further fast heating as seen in Table 2 and Figure 4. It is these samples which demonstrate a four-fold and higher increase in the TEC value, from 2.5 × 10^−6^ up to 10.5 × 10^−6^ K^−1^ within the temperature range from –50 to +50 °C, Figure 6. The TEC values in the temperature range of 250–350 °C revealed no variations for different treatments and were of 16 × 10^−6^ K^−1^. The absence of correlation between the degree of concentration heterogeneity of austenite and TEC value within the temperature range of 250–350 °C can be explained by the Curie point for the FeNi_31.1_ alloy (120 °C) being substantially exceeded. The concentration heterogeneity in Fe–Ni alloys as an atomic distribution variation relative to the state formed during homogenization with a further hardening can be achieved using electron irradiation. The Mössbauer results after 5.5 MeV electron irradiation with fluence of 5 × 10^18^ under 150 °C of the austenite from the FeNi_35_ alloy are shown in Figure 5. In terms of irradiation, the process of atomic redistribution proceeds in other way than the mechanism described above, namely, through the spinodal decomposition with homogeneous short-range order at the first stage with a further development of Fe_3_Ni and FeNi phases [6,18]. The FeNi_35_ alloy with the concentration heterogeneity formed after the electron irradiation demonstrated the TEC value growth from 0.6 × 10^−6^ to 6.5 × 10^−6^ K^−1^ within the temperature range from 20 to 100 °C. After annealing at 600–700 °C, the TEC values of tempered and irradiated samples appeared to be of the same order, ~1 × 10^−6^ K^−1^.

Thus, the way for development of the austenite with concentration heterogeneity using periodic polymorphous γ→α→γ transformation during the slow heating of the martensite-hardened Fe–Ni alloy of the invar range was suggested. In terms of the slow heating (at the rate of 0.2–0.4 K/min) an active Ni redistribution occurs on the developed borders between the initial α-martensite and appearing disperse γ-phase in accordance with the equilibrium diagram of Fe–Ni alloys [30,31]. Further fast heating from the temperature range of 400–500 °C up to 600 °C allows the α→γ transformation to be completed with retained austenite volume of about 40% from non-invar range. The variation of the concentration heterogeneity in the samples (volume and concentration different from the initial composition) provides regulation of austenite TEC.

## 4. Conclusions

Ni redistribution during the process of the polymorph α→γ phase transformation under the slow heating (0.2 K/min) was studied using Mössbauer spectroscopy for the case of the metastable FeNi_31.1_ alloy. The slow heating of quenched martensite up to 450–470 °C, with the following completion of α→γ transformation under the fast heating up to 600 °C in the salt bath, provides the development of about 40% of austenite with the non-Invar composition (13% with Ni content ~48 at.% and 27% with Ni content ≤ 29 at.%). Inhomogeneous nanosized regions are generated instead of plates of disperse Ni-enriched γ-phase and bordering structures of Ni-depleted α-martensite. The degree of concentration heterogeneity, given as a volume F of non-Invar regions, influences the TEC values. The volume variation of the non-Invar austenite part provides the ability to regulate the TEC value from 2.5 × 10^−6^ to 10.5 × 10^−6^ K^−1^ for the austenite with the concentration heterogeneity in the temperature range from –50 to 50 °C. The Mössbauer spectra shape, degree of concentration heterogeneity as well as TEC value for the Fe–Ni austenite were found to be similar to that of the FeNi_35_ Invar after the high-energy electron irradiation.

## Figures and Tables

**Figure 1 materials-15-08627-f001:**
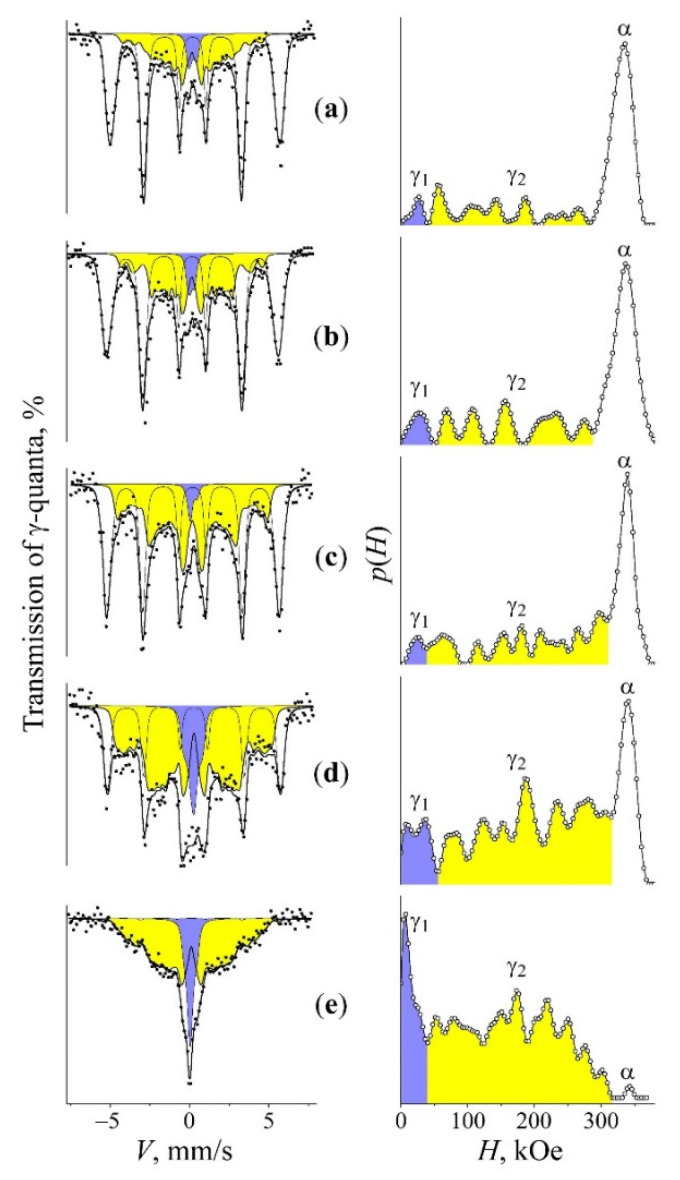
The Mössbauer spectra, *p*(*H*) distributions and the reconstituted spectral components γ_1_, γ_2_, and α of the FeNi_31.1_ alloy. Treatment: (**a**) tempering at 1050 °C, with subsequent quenching (i) in water and additional quenching (ii) using liquid nitrogen for the formation of α-martensite; (**b**–**e**) the treatment mentioned in (**a**) with the following α→γ transformation in the course of slow heating (0.2 K/min) up to 400, 470, 490, and 520 °C, respectively.

**Figure 2 materials-15-08627-f002:**
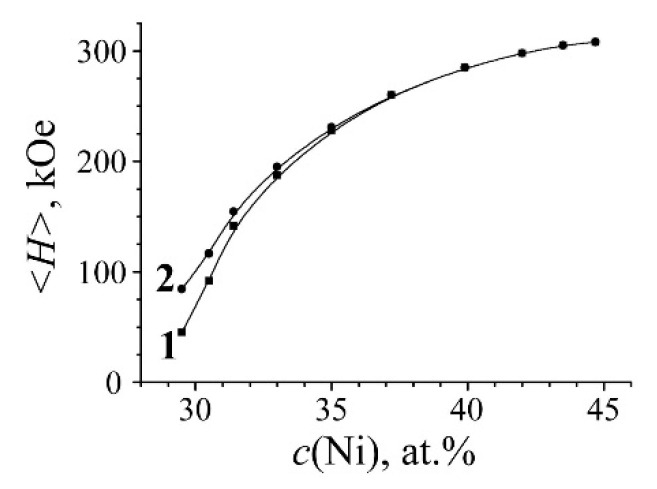
Plot of the mean effective magnetic field <*H*> and Ni content in the binary Fe–Ni alloys. Curve 1—<*H*> for the bulk spectrum; curve 2—<*H*> for the spectrum without the central component γ_1_.

**Figure 3 materials-15-08627-f003:**
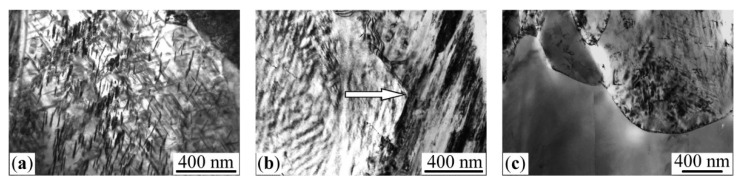
The structure of FeNi_31.1_ alloy formed during α→γ transformation under the heating at the rate of 0.2 K/min: (**a**) up to 430 °C; (**b**) up to 490 °C, an arrow indicates the growth direction of globular austenite; (**c**) up to 520 °C.

**Figure 4 materials-15-08627-f004:**
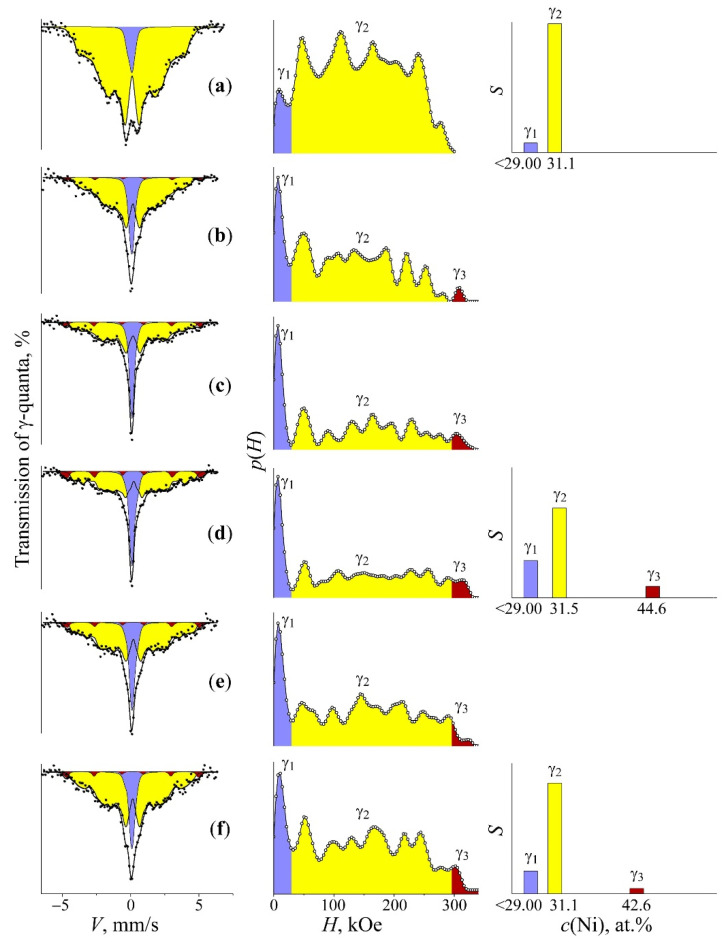
The Mössbauer spectra and the spectral components γ_1_, γ_2_, and γ_3_ reconstituted using *p*(*H*) and histograms of intensities *S* of Mössbauer spectra components and the concentration composition *c*(Ni) of the FeNi_31.1_ alloy. Treatment: (**a**) (i) quenching in liquid nitrogen for the formation of α-martensite and (ii) fast heating up to 600 °C for 10 min upon α→γ transformation; (**b**–**f**) slow heating (0.2 K/min) up to 400, 430, 470, 490, and 510 °C upon α→γ transformation, respectively, with subsequent fast heating up to 600 °C for 10 min upon austenitization.

**Figure 5 materials-15-08627-f005:**
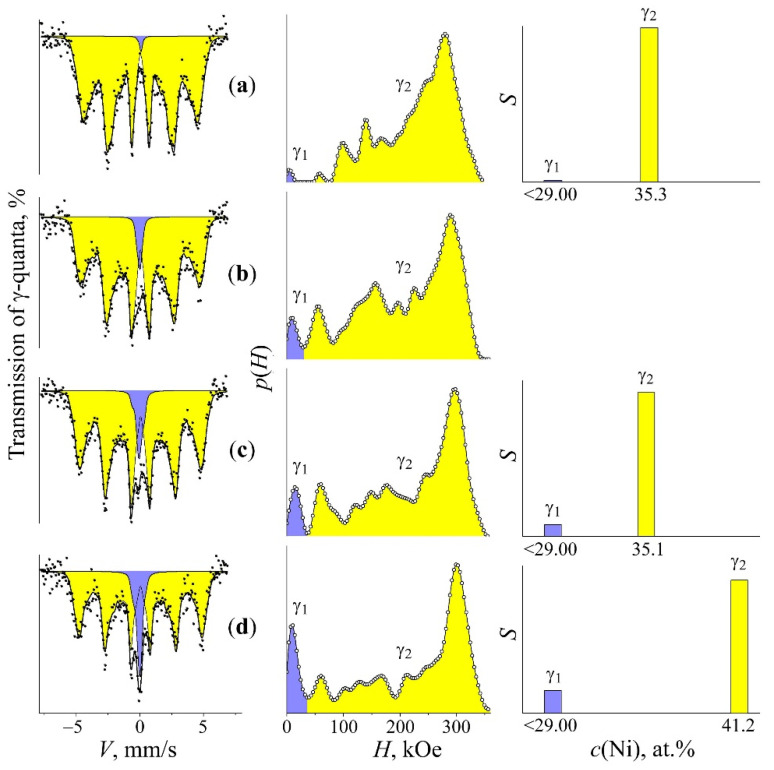
The Mössbauer spectra, the spectral components γ_1_ and γ_2_ reconstituted using *p*(*H*) and histograms of intensities *S* of Mössbauer spectra components and the concentration composition *c*(Ni) of the FeNi_31.1_ alloy after electron irradiation, with electron energy of 5 MeV. Treatment: (**a**) tempering at 1050 °C for austenite, with subsequent quenching in water; (**b**–**d**) treatment described in (**a**) with further electron irradiation at 120 °C, with fluence of 10^18^, 2 × 10^18^, and 5 × 10^18^ cm^−2^, respectively.

**Figure 6 materials-15-08627-f006:**
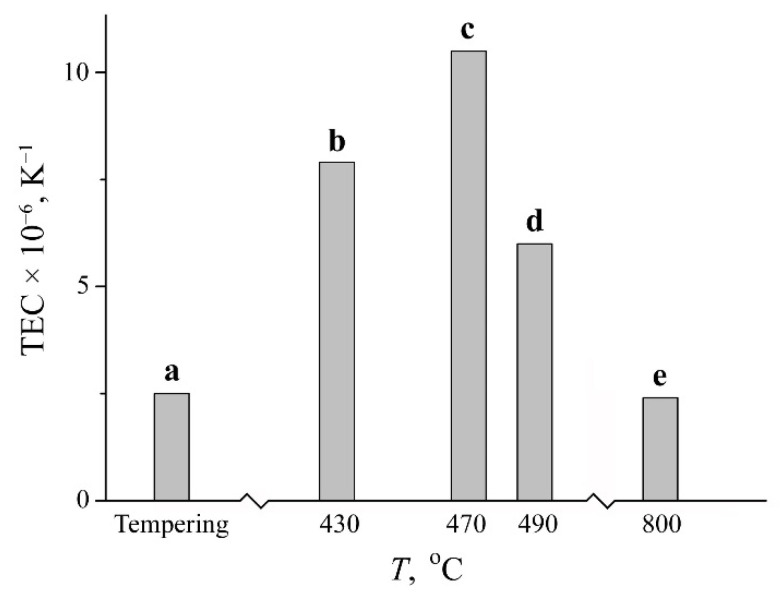
The thermal expansion coefficient values for the FeNi_31.1_ austenite with the concentration heterogeneity. Treatment: (**a**) tempering at 1050 °C for austenite, with subsequent quenching in water; (**b**–**d**) (i) quenching in liquid nitrogen for the formation of α-martensite and (ii) slow heating (0.2 K/min) up to 430, 470, and 490 °C upon α→γ transformation, respectively, with subsequent fast heating up to 600 °C for 10 min upon austenitization; (**e**) (i) quenching in liquid nitrogen for the formation of α-martensite and (ii) subsequent slow heating (0.2 K/min) up to 800 °C upon α→γ transformation.

**Table 1 materials-15-08627-t001:** Mean hyperfine magnetic field values <*H*> and integral intensities *S* of Mössbauer spectra components α, γ_1_ and γ_2_ (shown in Figure 1a–e) of the FeNi_31.1_ alloy after the slow heating of quenched martensite up to temperature *T*.

№	Heating Temperature *T*, °C	α-Phase		γ_1_-Phase	γ_2_-Phase	
		<*H*>_α_, kOe (±1)	*S*_α_, % (±5)	*S*_γ_, % (±2)	<*H*>_γ_, kOe (±5)	*S*_γ_, % (±5)
a	20 °C	331	64	4	144	32
b	400 °C	335	58	4	160	38
c	470 °C	337	47	3	184	50
d	490 °C	340	23	12	204	65
e	520 °C	344	1	21	166	78

**Table 2 materials-15-08627-t002:** Mean hyperfine magnetic field <*H*>, Ni concentration *c*(Ni) and integral intensities *S* for Mössbauer spectra components γ_1_, γ_2_ and γ_3_ (shown in Figure 4a–f) of the FeNi_31.1_ alloy after the slow heating of quenched martensite up to temperature *T* with a further fast heating up to 600 °C in the salt bath.

№	Heating Temperature *T*, °C	γ_1_-Phase	γ_2_-Phase			γ_3_-Phase		
		*S*_γ_, % (±2)	<*H*>_γ_, kOe (±5)	*c*(Ni)***, at.% (±0.1)	*S*_γ_, % (±3)	<*H*>_γ_, kOe (±3)	*c*(Ni)***, at.% (±0.1)	*S*_γ_, % (±1)
a	20 °C	7	144	31.1	93	–	–	0
b	400 °C	19	140	31.0	77	308	44.6	4
c	450 °C	27	147	31.3	64	306	43.8	9
d	470 °C	27	159	31.5	60	308	44.6	13
e	490 °C	20	150	31.3	71	304	43.2	9
f	520 °C	16	145	31.1	75	301	42.6	9

*c*(Ni)*******—mean effective concentration of Ni obtained using <*H*>_γ_ dependence shown in Figure 2.

## Data Availability

Not applicable.

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
