# Peer review of "Regulation of the Concentration Heterogeneity and Thermal Expansion Coefficient in the Metastable Invar FeNi31.1 Alloy"

_materials, 2022, doi:10.3390/ma15238627_

Round 1

Reviewer 1 Report

  In this paper, the metastable FeNi31.1 alloy was quenched in liquid nitrogen to obtain α phase structure. The αγ phase transformation was studied using Mössbauer spectroscopy. The concentration distribution of Ni in different phases during the phase transformation was analyzed. The experimental results confirm that the concentration heterogeneity of Ni in austenite and the existence of non-invar components in the alloy are the main reasons affecting the change of thermal expansion coefficient (TEC). The research results have a good reference value for the regulation of thermal expansion coefficient of invar alloy. After modification, it is agreed to accept and publish in this journal.

The main problems:

1. It is mentioned in the paper that the distribution of Ni element in the alloy is uneven, whether the EPMA method can be used for testing.

2. On lines 155 and 156, why does the same sentence appear twice.

3. Why is liquid nitrogen quenching used instead of other cooling medium?

Author Response

Response to REVIEWER

Dear Sir,

Thank you for your close peer-reviewing. We greatly appreciate your estimate of our work. The authors have taken into account all remarks and performed corrections, having marked these corrections in the text by a yellow color.

Q1:

“It is mentioned in the paper that the distribution of Ni element in the alloy is uneven, whether the EPMA (electron probe microanalyzer) method can be used for testing”.

A1:

In the case of austenite formation with concentration heterogeneity during the polymorphous  a®g transformation Ni-enriched regions of several nm in size are formed instead of disperse g-phase plates. If EPMA provides a possibility to examine the concentration heterogeneity with a gradient of content variation of some few atomic percent of Ni and use the nano-scale then EPMA can be applied, an example can be found in [V.V. Sagaradze, et.al Formation of low-temperature deformation-induced segregations of nickel in Fe–Ni-based austenitic alloys. Philosophical Magazine. 2020. V. 100, 2020, P. 1868—1879].

Q2:

“On lines 155 and 156, why does the same sentence appear twice”.

A2:

The required correction is done.  

Q3:

“Why is liquid nitrogen quenching used instead of other cooling medium?”.

A3:

The study is related to the periodic polymorphous phase transition g®a®g. One of the way to obtain a-martensite is a sample cooling down to the temperature lower than that of martensite phase transition g®a starting (Ms ~ –80 C) for FeNi31.1 In this case ~ 30 % of athermic lenticular martensite is developed in the sample. Other cooling medium can change a-phase morphology and volume.

Thank You once more for Your attention to the manuscript and Your comments.

Yours truly,

the authors

Reviewer 2 Report

In general, the writing in English should be improved with the help of an English-speaking professional.

It would be interesting if the authors could complete the structural characterization with X-Ray Diffraction and the magnetic characterization with VSM or SQUID Magnetometry to have another point of view. If it is too complicated for the authors, the paper could be published without them, but I would at least advise the authors to perform magnetometry characterization to measure the coercivity field and saturation magnetization of the samples.

The authors should describe the TEM characterization results in a self-consistent way and not only as a part of the discussion of the Mössbauer spectrometry results.

Lines 10-16: The abstract is insufficient. It must contain the main results.

Lines 12-14: “ The conditions of accelerated redistribution of Ni between the initial martensite α- 12 phase and newly formed γ-phase with a further austenite formation with the concentration hetero- 13 geneity and about 40 vol.% of non-invar alloy composition were established.”

This sentence is not well understood (there are 2 "with" in a row which makes it difficult to understand. Organise the information in this sentence into two sentences:

Lines 21-41: The introduction of the article is very limited. The authors should introduce and motivate some of the issues presented in this paper in a much more effective way.

For example, they should start by describing what is meant by invar effect and why it is magnetically interesting.

Make a state of the art of the FeNi phase diagram by pointing out the most important phases and their magnetic properties, the regions and aspects of the phase diagram that remain to be clarified......

In the context of the state of the art, the authors should indicate what they are going to contribute with this work, what synthesis and characterisation techniques they are going to use...

Linea 22-23: specify which magnetic phase transformation they are referring to.

Line 104: “3.2. Mechanical synthesis to produce an Fe–Cr–Ti–N ferrite-martensitic steels

This title must be wrong because up to line 104 the authors do not indicate that they are going to make Fe-Cr-Ti-N ferrite martensitic steels; they have only introduced the Fe-Ni alloy and all the results are for this alloy.

Author Response

Response to REVIEWER

Dear Sir,

Thank you for your close peer-reviewing. The authors have taken into account all remarks and performed corrections, having marked these corrections in the text by a yellow color.

Q1:

“In general, the writing in English should be improved with the help of an English-speaking professional”.

A1:

The paper was revised in accordance with recommendations of reviewer and English-speaking professional.  

Q2:

“It would be interesting if the authors could complete the structural characterization with X-Ray Diffraction and the magnetic characterization with VSM or SQUID Magnetometry to have another point of view. If it is too complicated for the authors, the paper could be published without them, but I would at least advise the authors to perform magnetometry characterization to measure the coercivity field and saturation magnetization of the samples”.

A2:

We would like to thank the reviewer for this advice. In future we will do our best to perform mentioned measurements of coercive force and saturation magnetization for further characterization of the invar material presented in the current paper.  

Q3:

“The authors should describe the TEM characterization results in a self-consistent way and not only as a part of the discussion of the Mössbauer spectrometry results”.

A3:

Current study was aimed to investigate the concentration heterogeneity in the Fe-Ni alloy formed via polymorphous g®a®g transitions and electron irradiation in relation to TEC. More detailed discussion of TEM data for polymorphous g®a®g in relation to Mössbauer spectroscopy was already performed in [19]. Thus, current paper presents new results of Mössbauer investigation in relation to electron microscopy.  

[19] Sagaradze, V.V.; Danilchenko, V.E.; L’Heritier, P.; Shabashov, V.A. The structure and properties of Fe–Ni alloys with a nanocrystalline austenite formed under different conditions of γ–α–γ transformations. Mat. Sci. Eng. A 2002, 337, 146–159. (https://doi.org/10.1016/S0921-5093(02)00023-0)

Q4:

“Lines 10-16: The abstract is insufficient. It must contain the main results”.

Q5:

“Lines 12-14: “ The conditions of accelerated redistribution of Ni between the initial martensite α- 12 phase and newly formed γ-phase with a further austenite formation with the concentration hetero- 13 geneity and about 40 vol.% of non-invar alloy composition were established.

This sentence is not well understood (there are 2 "with" in a row which makes it difficult to understand). Organise the information in this sentence into two sentences”.

A4, A5 :

Required corrections were performed.  

Q6:

“Lines 21-41: The introduction of the article is very limited. The authors should introduce and motivate some of the issues presented in this paper in a much more effective way.

For example, they should start by describing what is meant by invar effect and why it is magnetically interesting.

Make a state of the art of the FeNi phase diagram by pointing out the most important phases and their magnetic properties, the regions and aspects of the phase diagram that remain to be clarified......

In the context of the state of the art, the authors should indicate what they are going to contribute with this work, what synthesis and characterisation techniques they are going to use...”.

A6:

The introduction and conclusion parts were improved according to the reviewer remarks. 

Q7:

“Lines 22-23: specify which magnetic phase transformation they are referring to”.

A7:

Introduction was modified and required information as well as references were added (lines 27-30).

“Models associating the invar properties with concentration heterogeneities are built either on statistical fluctuations [10,11] or on local environment effects [6,12]. Nevertheless, both cases deal with magnetic inhomogeneities related to the composition fluctuations”.

Q8:

“Line 104: “3.2. Mechanical synthesis to produce an Fe–Cr–Ti–N ferrite-martensitic steels”. This title must be wrong because up to line 104 the authors do not indicate that they are going to make Fe-Cr-Ti-N ferrite martensitic steels; they have only introduced the Fe-Ni alloy and all the results are for this alloy”.

A8:

While preparing a draft manuscript using the required template the technical error occurred. Thank you for indication, it was corrected.

Thank You once more for Your close attention to the manuscript and Your comments.

Yours truly,

the authors

Reviewer 3 Report

firstly, the title is not suitable comparing to the presented content. According the presented results, the "Concentration Heterogeneity" is at microscopic and local scale, so it is necessary to precise this. The "Invar Fe–Ni Alloys" is not explicie in fact only the results from only one FeNi31.1 alloy has been presented in the manuscript.

secondly, the quantitative analysis protocole with Mössbauer study should be added and the associated precision/detectability/error should be discussed.

Thirdly, in figure 2, it is necessary to precise the origin of presented results (from bibliography ? from yoru own experiment ?) and discuss the corresponding precision;

- forthly, for results presented in table 1 and 2, it is necessary to add the corresponding precision/error, it is absolutely necessary to explain what correspond to the parameter S;

- fifthly, in figure 4 and 5, it is necessary to explain what correspond to the S parameter; 

-sixthly, it is necessary to assocaited an error bar with each numerical value

- seventhly, before the conclusion part, it is extremely important to add a separated discussion part to leave a global synthesis about the different results and to explain the temperature effect in relationship with the local Concentration Heterogeneity. This discussion is absolutely necessary.

Author Response

Response to REVIEWER

Dear Sir,

Thank you for your close peer-reviewing. The authors have taken into account all remarks and performed corrections, having marked these corrections in the text by a yellow color.

Q1:

“The title is not suitable comparing to the presented content. According the presented results, the "Concentration Heterogeneity" is at microscopic and local scale, so it is necessary to precise this. The "Invar Fe–Ni Alloys" is not explicie in fact only the results from only one FeNi31.1 alloy has been presented in the manuscript”

A1:

The title was corrected with indication of FeNi31.1 only. Electron microscopy was applied to study g-phase regions with uneven concentration (evaluation of the shape and appearance of contrasts comparable with g -plates of a few nm thickness). In the case of electron irradiation of the FeNi31.1 alloy the process of atomic redistribution goes as spinodal decomposition [6].  

Q2:

“The quantitative analysis protocole with Mössbauer study should be added and the associated precision/detectability/error should be discussed”.

Q3:

“In figure 2, it is necessary to precise the origin of presented results (from bibliography ? from your own experiment ?) and discuss the corresponding precision”.

Q4:

“For results presented in table 1 and 2, it is necessary to add the corresponding precision/error”.

Q6:

“It is necessary to assocaited an error bar with each numerical value”.

A2, A3, A4, A6:

Tables 1 and 2 illustrate Mössbauer data, namely, mean effective fields and integral intensities S of subspectra related to the corresponding phases. Effective Ni concentration in g-phase was evaluated via mean effective field value obtained as a centroid of p(H) distribution for Fe-Ni austenite in the composition region of 29.5..43.5 at.% (See Figure 2). These curves were obtained by us in [25, 29]. The errors of Mössbauer parameters’ values are given in Figure 2 and Tables 1 and 2.

Q4:

“It is absolutely necessary to explain what correspond to the parameter S”

Q5:

“In figure 4 and 5, it is necessary to explain what correspond to the S parameter”.

A4, A5:

S is related to the integral intensity of subspectra associated with a correspondent phase. Required information was added to the captions of Figures 4 and 5, Tables 1 and 2. Histograms shows S in arbitrary units.

Q7:

“Before the conclusion part, it is extremely important to add a separated discussion part to leave a global synthesis about the different results and to explain the temperature effect in relationship with the local Concentration Heterogeneity. This discussion is absolutely necessary”

A7:

Additional information about the concentration heterogeneity formation by means of periodic polymorphous transformation g®a®g as well as comparison with concentration heterogeneity of austenite formed via spinodal decomposition under the high-energy electron irradiation were added to the introduction and the final discussions.  

The conclusion about different nature of the concentration heterogeneity in both cases was suggested. However, both allow obtaining austenite with concentration heterogeneity and regulating the TEC.

Thank You once more for Your attention to the manuscript and Your comments.

Yours truly,

the authors

Round 2

Reviewer 3 Report

in the revised version of the manuscript, author has made necessary corrections including most of remarks from reviewers.